# The Role of PTEN in Epithelial–Mesenchymal Transition

**DOI:** 10.3390/cancers14153786

**Published:** 2022-08-03

**Authors:** Olga Fedorova, Sergey Parfenyev, Alexandra Daks, Oleg Shuvalov, Nickolai A. Barlev

**Affiliations:** Institute of Cytology RAS, 194064 St. Petersburg, Russia; gen21eration@gmail.com (S.P.); alexandra.daks@gmail.com (A.D.); oleg8988@mail.ru (O.S.)

**Keywords:** association of PTEN expression and EMT, molecular crosstalk between EMT-controlling transcription factors and PTEN, signaling networks of PTEN and EMT, regulation of miRNAs, lncRNAs and PTEN

## Abstract

**Simple Summary:**

The PTEN phosphatase is a ubiquitously expressed tumor suppressor, which inhibits the PI3K/AKT pathway in the cell. The PI3K/AKT pathway is considered to be one of the main signaling pathways that drives the proliferation of cancer cells. Furthermore, the same pathway controls the epithelial–mesenchymal transition (EMT). EMT is an evolutionarily conserved developmental program, which, upon aberrant reactivation, is also involved in the formation of cancer metastases. Importantly, metastasis is the leading cause of cancer-associated deaths. In this review, we discuss the literature data that highlight the role of PTEN in EMT. Based on this knowledge, we speculate about new possible strategies for cancer treatment.

**Abstract:**

Phosphatase and Tensin Homolog deleted on Chromosome 10 (PTEN) is one of the critical tumor suppressor genes and the main negative regulator of the PI3K pathway. PTEN is frequently found to be inactivated, either partially or fully, in various malignancies. The PI3K/AKT pathway is considered to be one of the main signaling cues that drives the proliferation of cells. Perhaps it is not surprising, then, that this pathway is hyperactivated in highly proliferative tumors. Importantly, the PI3K/AKT pathway also coordinates the epithelial–mesenchymal transition (EMT), which is pivotal for the initiation of metastases and hence is regarded as an attractive target for the treatment of metastatic cancer. It was shown that PTEN suppresses EMT, although the exact mechanism of this effect is still not fully understood. This review is an attempt to systematize the published information on the role of PTEN in the development of malignant tumors, with a main focus on the regulation of the PI3K/AKT pathway in EMT.

## 1. Introduction

Metastasis is a biological event during which the cellular activities of migration, survival and proliferation are tightly coordinated to ensure that cancer cells leaving the primary tumor location are able to set up a new colony in another metastatic niche. The epithelial–mesenchymal transition (EMT) is a process by which epithelial cells lose their apicobasal polarity and intercellular contacts, transforming into mesenchymal cells, and thus acquiring the ability to migrate and invade. The EMT program is instrumental to such biological processes as embryonic development and metastasis [1]. The main events associated with EMT are the downregulation of E-cadherin and zona occludens 1 (ZO-1) as epithelial markers, and the induction of mesenchymal markers such as vimentin, N/P-cadherins, and β-catenin. These are achieved by a fine-tuned network of transcription factors and signaling pathways [1,2]. The core set of EMT master regulators includes zinc-finger E-box-binding homeobox 1 (ZEB1) and ZEB2, Snail (SNAI1), Slug (SNAI2), and Twist-related protein 1 (TWIST1) [3].

Upon the loss of E-cadherin, epithelial cells in the primary carcinoma lose their cell–cell adhesion properties and focal adhesion contacts with the extra cellular matrix (ECM), resulting in loss of epithelial polarity. However, it should be noted that the majority of cancer cells from the primary tumor acquire an intermediate state of EMT or pseudo-EMT that is characterized by both epithelial and mesenchymal phenotypes [4]. Due to the intrinsic heterogeneity of tumors, only several individual cells or groups of cells are able to detach from the primary tumor and intravasate into the bloodstream to subsequently form metastases. Tumor cells extravasate from blood vessels near specific metastatic niches and undergo the reverse process, called mesenchymal–epithelial transition (MET). This is an important step for the colonization of new tumor niches and, ultimately, for the formation of metastasis, since MET causes cells to proliferate. It is important to mention that EMT in cancer cells is not only reversible but can be stopped at any stage [5].

Recent studies have shown that EMT also plays an important role in the acquisition of drug resistance by cancer cells [6,7] as well as resistance to radiotherapy [8,9]. EMT-induced drug resistance is mediated by attenuating the proliferation rate, upregulating drug efflux, blocking the apoptotic mechanisms of cancer cells, and by increasing evasion from immune surveillance [10,11,12].

Furthermore, activation of the EMT program may also trigger the generation of cancer stem cells (CSCs), which are suggested to be another mechanism of drug resistance. CSCs, also known as tumor-initiating cells (TICs), comprise a class of pseudo-multipotent cells that are able to transdifferentiate into different subtypes within a tumor-initiating tissue [13,14].

PTEN is the main negative regulator of the PI3K/AKT pathway, and hence is considered to be one of the major tumor suppressors. Thus, it is not surprising that the loss of its expression or functions is observed in different cancers [15]. The PI3K/AKT signaling pathway plays an important role in carcinogenesis by orchestrating a number of cellular processes including proliferation, differentiation, metabolism, and migration. The PI3K/AKT pathway is intimately intertwined with EMT at multiple levels.

A large number of studies have shown that PTEN negatively affects EMT. However, the exact role of PTEN in metastasis remains unclear. In this review we discuss the role of PTEN in metastasis, including crosstalk between the signaling networks of PTEN and EMT, which ultimately affects the gene expression programs of cellular plasticity and metastasis.

## 2. PI3K Pathway and PTEN

### 2.1. The PI3K Pathway

PI3K pathway signaling is activated by cell surface located growth factor receptors (RTKs), G protein-coupled receptors (GPCRs), cytokine receptors, and integrins [16]. RTKs are a family of high-affinity cell surface receptors that sense growth factors, hormones, cytokines, and other signaling molecules, and convert these signals into the phosphorylation cascade inside the cell. Over 58 gene encoding RTKs have been identified in the human genome [17].

The key element of the PI3K/AKT signaling pathway, PI3K, is a member of the family of enzymes that phosphorylate phosphatidylinositol at the 3′-position of the inositol ring [18]. The PI3K family members can be divided into three classes depending on their structural organization and substrate specificity. The most important role in the formation of cancerous tumors is type I PI3K. They are represented by heterodimers consisting of the catalytic subunit p110 (p110α, p110β, p110γ, and p110δ) and one of the regulatory subunit p85α, p85β, p55α, p55γ, p50α, p101, and p87 [19]. Typically, the catalytic subunit (p110) binds to the regulatory subunit to stabilize the PI3K heterodimer, and thus inhibits PI3K activation. The p85 regulatory subunit of PI3K contains SH2-domains (Src homology-2) that mediate binding to adaptor proteins or tyrosine receptors to facilitate localization and activation of PI3K at the membrane [20]. There is less information about PI3K types II and III; however, it is known that PI3K II consists only of catalytic subunits PI3KC2α, PI3KC2β, or PI3KC2γ, in contrast to class III, which exists in a heterodimeric form and consists of the catalytic subunit Vps34 and the regulatory protein p150 [21].

Activation of type I PI3K occurs through distinct cellular responses downstream of various receptors, including receptor tyrosine kinases (RTKs), cytokine receptors, B-cell antigen receptors, some integrins, and a subset of G protein-coupled receptors. Activation of the receptors upon binding the ligand leads to the autophosphorylation of tyrosine residues within the RTK, and PI3K is then recruited to the membrane by direct binding to consensus phosphotyrosine residues located in the C-terminus of the receptor via one of the two SH2-domains in the PI3K regulatory subunit. This leads to the activation of the catalytic PI3K subunit, resulting in the formation of a secondary messenger phosphatidylinositol (3,4,5)-trisphosphate (PIP3) from the phosphatidylinositol (4,5)-bisphosphate (PIP2) substrate. PIP3 then recruits signaling proteins to the inner membrane that contain pleckstrin domains (PH), including 3′-phosphoinositide-dependent kinase 1 (PDK1) and protein kinase B (AKT). PH-domains are the conserved domains found in hundreds of mammalian proteins involved in intracellular signaling and are responsible for recruiting proteins to lipid membranes [22].

The most important mediator of the PI3K pathway is protein kinase AKT [23]. There are three isoforms of AKT: AKT1, AKT2, and AKT3 [24,25,26]. These isoforms have a common conserved domain structure, consisting of the N-terminal domain (pleckstrin homology; PH-domain), the kinase domain, and the C-terminal regulatory domain containing a hydrophobic motif [26]. It was shown that AKT1 is expressed ubiquitously, while AKT2 is expressed in insulin-sensitive tissues, and AKT3 is expressed in the brain and testes [27]. PIP3 can bind the PH-domain of AKT and induce the translocation of AKT to membranes, where AKT is phosphorylated at two sites: Thr308 and Ser473, which, in turn, leads to the activation of the enzyme [28,29,30]. PDK1 is responsible for the phosphorylation of Thr308 [31], and the phosphorylation of Ser473, which can be carried out by integrin-linked kinase (ILK) or the mTORC2 protein complex, PKCε [32,33,34]. According to Ebner et al. [35], active AKT is predominantly associated with cellular membranes and is critically dependent on the engagement with PIP3. Accordingly, the transforming mutation D323A, which uncouples kinase activation from PIP3 binding, leads to the accumulation of hyperactive phosphorylated AKT in the cytosol [35].

The activation and stabilization of AKT is regulated by phosphorylation not only at the two major sites (Thr308 and Ser473) but also at many additional sites. It was shown that AKT1 has 31 potential phosphorylation sites, while AKT2 possesses 22, and AKT3 has 18 phosphorylation sites [36]. For example, AKT can be phosphorylated by CK2 at Ser129, which augments its catalytic activity [37]. Liu et al. [38] found that phosphorylation at S477/T479 was essential for AKT activation.

The activated form of AKT promotes cell survival by suppressing the expression of genes responsible for cell death, or by activating the expression of genes that contribute to survival [39,40]. Thus, AKT has been shown to inhibit the activity of the following: Foxo family members FKHR (FoxO1), FHHRL1 (FoxO3), and AFX (FoxO4); Bad, which is a pro-apoptotic member of the Bcl-2 protein family; kinase ASK1, which activates JNK and p38 MAP kinases; and procaspase-9 which plays an important role in the signaling pathway of apoptosis.

The Forkhead (FoxO) family of transcription factors includes FKHR/FoxO1, FoxO2, FKHRL1/FoxO3, and AFX/FoxO4 [41]. Forkhead proteins are important downstream targets for AKT, and their phosphorylation by AKT can regulate cell survival by changing the expression of their target genes [42]. In addition, these transcription factors can inhibit cell proliferation by suppressing cyclin D, and can induce apoptosis by transactivating the Fas ligand [43,44].

PI3K/AKT/mTOR/p70S6K is one of the main signaling pathways that regulate autophagy under stress conditions, for example, during fasting, oxidative stress, or infection [45]. It has been shown that GSK3β is responsible for the induction of apoptosis upon inhibition of AKT. Suppression of GSK3β also decreases the level of basal apoptosis and induces proliferation [46].

The protein kinase mTOR (also called the mammalian target of rapamycin) is an evolutionarily conserved protein from yeast to humans. The mTOR activity is finely controlled both positively and negatively by upstream regulators. Positive regulators of mTOR signaling include growth factors and their receptors, such as insulin-like growth factor-1 (IGF-1) and its cognate receptor IFGR-1, members of the human epidermal growth factor receptor (HER) family and their associated ligands, as well as vascular endothelial growth factor receptors (VEGFR) and their ligands that signal mTOR via PI3K/AKT [47]. mTOR exists in the cell as a subunit of the intracellular multimolecular signaling complexes TORC1 and TORC2 [48]. The mTORC1 complex consists of mTOR, Raptor, mLST8, and PRAS40. Importantly, it was found to be sensitive to rapamycin. This discovery allowed to develop the first generation of mTOR inhibitors. The mTORC2 complex consists of mTOR, Rictor, Sin1, and mLST8. It is less sensitive to rapamycin, and its role in normal cell function and tumorigenesis is not fully understood [49]. It is important to underline that mTORC2 activates AKT, thereby promoting cell proliferation and survival. The canonical mTOR activation pathway depends on PI3K/AKT signaling, but alternative non-AKT-dependent activation pathways are also known, such as the Ras/MEK/ERK pathway [50] (Figure 1).

### 2.2. PTEN

Phosphatase and Tensin Homolog deleted on Chromosome 10 (PTEN) is a tumor suppressor that exhibits phosphatase activity preferentially against members of the PI3K oncogenic pathway. Thus, PTEN is considered to be a major negative regulator of the PI3K/AKT pathway. Oppositely, PTEN inactivation augments the activity of the PI3K/AKT pathway, resulting in the better survival, proliferation, differentiation, and migration of cancer cells.

Under normal conditions, only a small fraction of PTEN molecules bind to the plasma membrane and the rest mainly localize in the cytoplasm. To perform its membrane-bound functions, PTEN first needs to be activated through post-translational modifications and then recruited to the plasma membrane. PTEN acts as a lipid phosphatase that removes the phosphate group from the second messenger PIP3 (phosphatidylinositol (3,4,5)-trisphosphate) to generate PIP2 (phosphatidylinositol (4,5)-bisphosphate), thereby controlling cellular proliferation, survival, growth, and migration [51]. Notably, PTEN acts not only through the lipid but also via its protein phosphatase activity. PTEN is considered to be a dual protein and lipid phosphatase. Furthermore, PTEN can operate non-enzymatically [52].

The regulation of PTEN expression occurs at multiple levels, including epigenetic silencing, transcriptional repression, post-transcriptional regulation by miRNAs, and disruption of competitive endogenous RNA (ceRNA) networks. In addition, PTEN-interacting proteins also affect the activity and functions of PTEN [53,54].

Not surprisingly, PTEN itself, or its functions, are frequently lost either partially or fully in different types of cancer [55]. PTEN inactivation can occur through various genetic alterations such as epigenetic mechanisms (promoter hypermethylation), large chromosomal deletions, or point mutations [56].

PTEN activity can also be regulated by various signaling cues. For example, the sphingosine 1-phosphate2 receptor (S1P2R), which belongs to the S1PR family of G protein-coupled receptors (GPCR), has been shown to activate PTENs by coupling them to the Rho-dependent activation. As a result, the PI-3 kinase pathway is inhibited and cell migration is attenuated [57].

Lima-Fernandes et al. [58] used an elegant in cellulo approach to monitor the conformational state of PTEN in response to various extracellular signals. By employing the intramolecular bioluminescence resonance energy transfer (BRET)-based PTEN conformation-specific biosensor as a reporter, they uncovered that PTEN was activated by several G protein-coupled receptors, including the thromboxane A2 receptor (TPαR) and the muscarinic M1 receptor (M1MR), which were not previously recognized as PTEN regulators.

As mentioned above, PTEN localizes both to the cytoplasm and nucleus. At the plasma membrane, PTEN plays an important role in activating apoptosis by suppressing AKT and, as a result, activating caspase-8 and caspase-9, increasing expression of several pro-apoptotic genes [59].

Localization and activity of PTEN is regulated largely through post-translational modifications. PTEN monoubiquitination is implicated in the mediation of the nuclear accumulation of PTEN by several E3 ligases such as NEDD4, WWP2, XIAP, CHIP, and others, whereas, on the contrary, the polyubiquitination of PTEN induces its proteasome-mediated degradation [60,61,62,63,64,65]. SUMOylation of PTEN at lysine 254 was shown to be also involved in its nuclear translocation [66]. In this respect, it should be noted that Ca^2+^ mediates the interaction of PTEN with major vault protein (MVP) and regulates the localization of PTEN in the nucleus [67]. Furthermore, it was shown that PNUTS (protein phosphatase-1 nuclear targeting subunit, PPP1R10) interacts with PTEN, blocking the access of PTEN to membrane lipids and hence activating its translocation to the nucleus [68]. In turn, nuclear PTEN participates in DNA repair and replication, and gene expression, as well as controlling cell cycle progression [69]. Notably, members of the PI3K pathway, PI3K and AKT, are also found in the nucleus. Thus, PTEN can act as a lipid phosphatase both at the plasma and nuclear membranes [70,71].

Importantly, PTEN also regulates the stability and transcriptional activity of another tumor suppressor, p53 [72]. Moreover, PTEN was demonstrated to be a transcriptional target of p53 [73], thus forming a positive regulatory feedback loop. On the contrary, another transcriptional target of p53, E3-ubiquitin ligase MDM2, which is a negative regulator of p53, mediates its proteasome-mediated degradation [74]. Various inhibitors of the p53-Mdm2 interaction have been developed and are currently being investigated at different stages of clinical trials [75,76]. Interestingly, AKT can phosphorylate MDM2, which leads to the stabilization of the latter and thus keeps low levels of the tumor suppressor p53 [77]. In turn, inhibition of the PI3K/AKT pathway by PTEN mediates the accumulation of MDM2 in the cytoplasm [78] (Figure 2). Chung et al. [79] showed that, being localized in the nucleus, PTEN inhibits the expression of cyclin D1, thereby causing cell cycle arrest. Nuclear PTEN plays an important role in the maintenance of chromosomal integrity by association with the centromere specific binding protein C (CENP-C)—a key component of centromeres that is important for kinetochore attachment during mitosis [80,81]. Moreover, PTEN regulates the expression of and recruits Rad51 to chromatin favoring double strand break (DSB) repair [81]. Rad51 is one of the central figure factors of the DSB repair machinery that mediates homologous recombination repair [82]. In the nucleus, PTEN may act as regulator of DNA repair at different levels, and it is not surprising that cells lacking nuclear PTEN are sensitive to DNA damage. Importantly, the functioning of PTEN in the nucleus does not require its functional catalytic domain [66].

It is important to note that a number of mutations in the Tp53 gene convert p53 from a tumor suppressor to an oncogene. In this respect, Li et al. [83] showed that by increasing the level of mutant p53 (gain-of-function) via the inhibition of Mdm2-mediated degradation, PTEN can also be an oncogene. Thus, the mutational statuses of p53 and PTEN are important factors to be considered for understanding the aggressiveness of a tumor.

## 3. PTEN and EMT

### 3.1. PTEN and EMT Transcription Factors

The ZEB family of transcription factors consists of two members: ZEB1 (alternatively, TCF8 and δEF1) and ZEB2 (SIP1 and ZFXH1B). The members of this protein family interact with DNA through the binding of the two zinc-finger domains to E-boxes, and repress the expression of E-cadherin (Figure 3).

Liu et al. [84] showed that the repression of ZEB1 leads to the induction of PTEN. More detailed experiments revealed that ZEB1 affects the AKT pathway and inhibits the expression of PTEN in breast cancer cells. Moreover, ZEB1 binds to the PTEN promoter region in the MCF7 cell line. Using a luciferase reporter assay, Gu et al. [85] demonstrated that ZEB1 suppresses the transcription of the PTEN gene and as a result activates the AKT pathway.

ZEB2 is the transcription factor that, along with ZEB1, controls EMT. Interestingly, ZEB2 transcripts act as ceRNA towards miRNA targeting PTEN in melanoma cells. Moreover, the depletion of ZEB2 activates the AKT pathway and enhances oncogenic cell transformation of melanoma [86].

The SNAIL family of transcription repressors encompasses three members: SNAIL1 (SNAIL), SLUG (SNAIL2), and SNAIL3 (SMUC). Being important effectors of EMT, these proteins inhibit the expression of E-cadherin. In addition, the SNAIL family members were also shown to participate in PTEN regulation. It was demonstrated that SNAIL1 prevents p53 association with the PTEN promoter region after γ-radiation, thus downregulating the expression of PTEN [73,87]. Another member of this protein family, SLUG, was shown to repress PTEN by directly binding to the PTEN promoter in prostate cancer cells [88].

### 3.2. PTEN and Tumor Microenviroment

The EMT is triggered not only by intrinsic but also by extrinsic factors. Tumor cells interact with the tumor microenvironment (TME), including stromal cells, the extracellular matrix, and secretome elements (cytokines, growth factors, hormones, and so on). There is much evidence accumulated to date suggesting that hypoxia and reactive oxygen species (ROS) affect the onset of EMT. A common feature of various solid tumors is intra-tumoral hypoxia or a low level of oxygen. Hypoxia correlates with altered gene expression, inhibition of apoptosis, activation of autophagy, and EMT. Hypoxia also favors the production of ROS via several mechanisms [89,90,91]. Increased ROS levels have been detected in many types of cancers, correlating with tumor development and progression. Moreover, the reverse scenario is also true, i.e., that hypoxia can be mediated by increased levels of ROS [92], and the mitochondrial ROS is considered to be instrumental for hypoxia-induced EMT [93]. PTEN itself contains nucleophilic cysteine residues in the active site and thus is considered to be sensitive to oxidation. Cysteine residues (Cys-124 in the catalytic domain) can form an intramolecular disulfide bridge with Cys 71, inhibiting the phosphatase activity [94].

EMT is promoted by direct interaction between tumor and stromal cells. The tumor stroma consists of an extracellular matrix, immune and endothelial cells, as well as cancer-associated fibroblasts (CAFs). CAFs represent the major cellular component among all the stromal cells. Normal fibroblasts can become CAFs under the influence of cancer-derived cytokines. Inactivation of PTEN in the stroma correlates with poor outcomes of cancer patients [95]. For example, mutations in the PTEN gene of stromal fibroblasts were detected in mammary carcinoma [96]. Although these PTEN mutations identified in stromal cells might be due to contaminations by carcinoma cells or to EMT, downregulation of PTEN in CAFs might contribute to neoplastic progression. Trimboli et al. [97] showed that the inactivation of PTEN in stromal fibroblasts leads to the progression of mammary epithelial tumors. Moreover, PTEN-specific signatures in the tumor stroma of patients with breast cancer were identified by gene expression profiling. Thus, it is prudent to say that PTEN can be considered a critical tumor-suppressive component of stroma-specific signaling pathways in solid tumors.

### 3.3. TGF-β and PTEN

Transforming growth factor β (TGF-β) is a multifunctional cytokine family that regulates cell proliferation, apoptosis, differentiation and migration. This signaling protein family is considered to be one of the key regulators of the EMT process. The TGF-β family in mammals includes three members, TGF-β1, TGF-β2, and TGF-β3. TGF-βs are expressed as pro-proteins, and furin proteases are necessary for proteolytic processing and the activation of TGF-βs. Activated TGF-β complexes bind to serine/threonine kinases-TGF-β receptors [98]. There are two signaling pathways triggered by TGF-β: the canonical pathway induces the phosphorylation of the SMAD family of transcription factors, while the non-canonical pathway includes MAPK, PI3K/AKT, and Rho GTPase. Upon phosphorylation, SMADs form a trimeric structure (SMAD2, SMAD3, and SMAD4) and translocate to the nucleus to regulate the transcription of target genes by directly binding to DNA [99] (Figure 4).

The regulation of PTEN expression by TGF-β signaling is multifaceted and depends on the cellular context. While PTEN was initially identified as a TGF-β-regulated and epithelial cell-enriched phosphatase (TEP1) [100], this concept was challenged by Kimbrough-Allah et al. [101], who reported that, in prostate cancer cells, TGF-β increased the protein level of PTEN, but had no effect on the PTEN mRNA expression level. However, it should be noted that the effect of TGF-β on the protein level of PTEN was very modest and is unlikely to be physiologically relevant. In turn, PTEN overexpression decreased AKT phosphorylation induced by TGF-β [101].

In contrast to the report of Kimbrough-Allah et al., in SMAD4-null pancreatic cancer cells, it was shown that TGF-β suppressed PTEN expression. It is important to note that the PTEN-coding gene is rarely mutated in pancreatic cancers and the authors hypothesized that PTEN regulation by TGF-β can confer changes in cell growth [102].

Furthermore, in SMAD4-null colon cancer cells, TGFβ also downregulated PTEN mRNA and concomitantly induced growth proliferation [103]. Collectively, the literature data strongly suggest that, in most cancer types, TGF-β attenuates PTEN expression.

Interplay between PI3K/AKT and TGF-β/SMAD signaling pathways was identified in wild-type mouse endometrial epithelial cells. Eritja et al. [104] demonstrated that TGF-β induced apoptosis via the activation of PTEN transcription and the inhibition of the PI3K/AKT pathway. Additionally, it was shown that both the expression of PTEN and AKT phosphorylation depend on SMAD3 activity. Thus, in this study, the link between AKT/PTEN/SMAD3 in response to TGF-β was demonstrated [104].

PTEN activity depends on various post-translational modifications such as oxidation, acetylation, and ubiquitination [105]. For example, phosphorylation at sites Ser380, Thr382, and Thr383 leads to the suppression of PTEN phosphatase activity [106]. Aoyama et al. [107] demonstrated that the activation of TGF-β increased the phosphorylation of PTEN at these residues. Moreover, the expression of PTEN with mutations in these inhibitory phosphorylation sites was able to repress EMT induced by TGF-β by blocking β-catenin translocation from E-cadherin complexes at the cell membrane into the cytoplasm [107].

Indeed, β-catenin plays the central role in the canonical Wnt/β-catenin signaling pathway, which regulates embryonic development and, in the event of abnormal activation, leads to metastases. There is a crosstalk between TGF-β and Wnt/β-catenin signaling pathways [108,109]. β-catenin is involved in various processes such as the regulation of cell adhesion and the transcription of genes. E-cadherin binds to β-catenin and promotes cell–cell adhesion. The activation of TGF-β induces the dissociation and stabilization of β-catenin in the cytoplasm, allowing β-catenin to be translocated to the nucleus. β-catenin is a critical element of Wnt signaling. Nuclear β-catenin reflects the aberrant activation of Wnt/β-catenin signaling in cancer cells, leading to dysregulated transcription and hence the activation of EMT [110,111].

### 3.4. NOTCH and PTEN

The Notch signaling pathway is transmitted through evolutionarily conserved Notch receptors, which are localized on the plasma membrane and affect the transcription of different genes. There are four isoforms of Notch proteins: Notch1, 2, 3, and 4, which work as heterodimeric receptors by binding two classes of ligands: Delta-like proteins and Jagged. The binding of ligands allows proteolytic cleavage to release active Notch intracellular domains (NICD). Active NICD translocate to the nucleus and bind transcription factors to regulate gene expression. The activity of Notch receptors is regulated not only by binding to ligands but also by post-translational modifications, epigenetic factors, and ligand cleavage [112]. Notch pathways take part in numerous embryonic developmental processes and are implicated in cancer progression and EMT (Figure 4). Saad et al. [113] demonstrated that the overexpression of Notch1 increases Snail1 expression and decreases the level of E-cadherin. Indeed, the Notch pathway regulates EMT in different types of cancer. For example, the activation of Notch1 enhanced EMT in gefitinib-resistant lung cancer cell lines. On the contrary, the inhibition of Notch1 reversed the mesenchymal phenotype of cancer cells to the epithelial one and increased the sensitivity of cancer cells to gefitinib [114]. At the same time, Notch3 has the opposite effect on EMT through the transactivation of Hippo/Yap signaling in breast cancer [115]. Zhang et al. [116] showed that the PTEN gene was transactivated by Notch3 in breast cancer cells, resulting in the inhibition of their migration and proliferation, thereby contributing to a favorable prognosis for breast cancer patients.

Some of the earlier proofs of the link between PTEN and EMT came from the study demonstrating that PTEN inhibits cell migration through the phosphatase-mediated dephosphorylation of the FAK (focal adhesion kinase) protein [117]. FAK is a cytoplasmic tyrosine kinase that plays an important role in EMT and mediates the signaling of growth factor receptors and integrins. Moreover, FAK induces a PI3K signaling pathway and regulates the transcriptions of vimentin and Snail. Thus, Li et al. [118] demonstrated that Notch1 signaling regulates the expression of PTEN, EMT-associated genes, and the activation of FAK through the transcriptional regulator RBP-Jκ in tongue cancer. The study by Baker et al. [119] confirmed that Notch1 downregulates PTEN expression leading to the inhibition of ERK1/2 signaling in HER2-positive breast cancer.

### 3.5. NF-κB and PTEN

NF-κB (Nuclear factor-κB) represents several inducible transcription factors, which regulate the transcription of genes involved in the immune response, survival, angiogenesis, and EMT. The NFκB family consists of five subunits: Rel (cRel), p65 (RelA, NFκB3), RelB, p50 (NFκB1), and p52 (NFκB2). These proteins contain the Rel-homology domain (RHD), which is responsible for homo- or hetero-dimerization, nuclear localization, and transcription [120]. Multiple links between the EMT and NFκB pathway have been demonstrated (Figure 4).

By using a luciferase reporter assay and chromatin immunoprecipitation, Pires et al. [121] showed that NF-κB/p65 directly binds promoters of the SLUG, TWIST1, and SIP1 genes in breast cancer cells. Several lines of evidence point to the link between NF-κB and EMT. The correlation between NF-κB expression and EMT-specific transcription factors was shown in breast cancer, renal carcinoma, prostate cancer, and other types of cancer [122,123,124]. Julien et al. [125] demonstrated that the AKT-dependent activation of NF-κB induces the expression of Snail, which in turn inhibits the expression of PTEN, thereby forming the PTEN/Akt/NF-κB/Snail feedback loop. Moreover, it was shown that the A-kinase-interacting protein 1 (AKIP1) induced NF-κB-mediated EMT by downregulating PTEN in cervical cancer [126].

Elegant experiments performed by Zhao [127] and colleagues established the link between PTEN and the chromatin-mediated regulation of the NF-κB pathway in prostate and breast cancers. Decreased expression of PTEN resulted in the stabilization of the chromatin helicase DNA-binding factor CHD1, which engages the trimethyl lysine-4 histone H3 modification to activate the transcription of cancer-promoting genes involved in the NF-κB pathway [127].

### 3.6. Integrins and PTEN

Integrins represent a family of transmembrane receptors that mediate cell adhesion to extracellular matrices. These receptors are presented by heterodimers composed of 18α and 8β subunits, which form at least 24 different integrin heterodimers. The general structure of α and β subunits appears to be similar. These subunits possess a globular N-terminally located extracellular domain, the transmembrane domain, and the small C-terminal cytoplasmic tail [128]. The α-subunits mediate ligand specificity, and the β subunits form a bridge between the extracellular matrix and integrins. These receptors control a variety of signaling pathways including actin polymerization, nucleation, PI3K, AKT, and MAPK [129]. Being key regulators of cellular adhesion, integrins also participate in the EMT process. Integrins support the colonization of metastatic cells and mediate the survival of circulating tumor cells [128]. Integrins act as receptors and transduce signals through additional proteins that connect to the cytoskeleton, transmembrane growth factor receptors, and cytoplasmic kinases [130]. Integrins themselves lack any enzymatic activity and all of the functions they exert are mediated via protein–protein interactions, resulting in the assembly of actin filaments [131]. The clustering of integrins and the reorganization of actin through adaptor proteins enhances the formation of stable aggregates named focal adhesion sites (FAS) [130]. These FASs mediate signal transduction to ensure different aspects of the EMT process such as proliferation, wound healing, and migration [132] (Figure 4).

Using several lung cancer cell lines with an ectopic expression of PTEN, Yu et al. identified differentially expressed genes by microarray analysis. One of the genes on top of the microarray list was αVβ6 integrin. The expression of PTEN decreased the protein level of αVβ6 integrin, thereby affecting the migration and invasion of lung cancer cell lines. As mentioned above, one of the earliest studies elucidating the impact of PTEN on the extracellular matrix (ECM) rearrangement during EMT was performed by Tamura et al. [133]. Importantly, PTEN-mediated dephosphorylation of FAK at Tyr397 is a key event ensuring the formation of intercellular contacts mediated by integrins. PTEN therefore negatively regulates cell interactions with the extracellular matrix [134]. Interestingly, on the other hand, PTEN is a substrate of FAK-mediated Tyr336 phosphorylation. This event, in turn, leads to increased phosphatase activity, an augmented protein–lipid interaction, and the stability of PTEN [135]. Thus, there is a crosstalk between the activity of FAK and the stability of PTEN.

Moreover, dephosphorylation by PTEN affects FAK adhesion to collagens. Collagens are the major fibrous proteins of ECM [136], while paxillin is a multidomain protein that transduces signals from integrins in order to reorganize the actin cytoskeleton. It also modulates actin filament dynamics and mediates the engagement of integrins with extracellular matrices [137]. It was demonstrated that the interactions of PTEN with FAK and paxillin weaken their interactions with ECM components, e.g., collagens I/IV [138]. Thus, PTEN may contribute to the process of destabilization of cell adhesion to ECM.

### 3.7. PTEN and Some Scaffolding Proteins

As mentioned above, one of the main hallmarks of EMT is the loss of epithelial markers, i.e., E-cadherin. The lipid phosphatase activity of PTEN is instrumental in stabilizing E-cadherin on the protein level. The expression of wild-type PTEN suppressed the invasive phenotype in an E-cadherin-dependent manner. Thus, the inactivation of PTEN can activate cell dissemination and metastasis [139]. While the extracellular region of E-cadherin extends from the cell surface and is involved in cell adhesion, the intracellular region interacts with catenins to form a complex E-cadherin/catenin. E-cadherin/catenin protein complexes are considered to be a part of bigger complexes that involve growth factor receptors and scaffolding molecules. For example, MAGI-1b, which belongs to the group of membrane-associated guanylate kinases (MAGUK), interacts with β-catenin. Kotelevets et al. showed that PTEN interacted with β-catenin by binding MAGI-1b, a scaffold protein with PDZ domains that mediate multiple protein–protein interactions [140]. Moreover, the expression of MAGI-1b is important for the interaction of junction complexes and PTEN, which promotes E-cadherin-dependent cell aggregation, thus suppressing invasion [140]. Interestingly, MAGI1 interacts with Thyroid Hormone Receptor Interactor 6 (TRIP6). In turn, enhanced levels of cytoplasmic TRIP6 compete away β-catenin from MAGI1. Thus, TRIP6 allows β-catenin translocation to the nucleus and subsequently activates Wnt/β-catenin signaling [141]. In addition, TRIP6 favors the dissociation of PTEN from the E-cadherin complexes, thereby activating invasiveness [142]. Thus, scaffold proteins play an important part in the participation of PTEN in EMT, affecting both protein–protein interactions and signaling pathways.

Furthermore, PTEN interacts with a scaffolding protein β-arrestin1, which regulates the signal transduction mediated by G protein-coupled receptor (GPCR). β-Arrestin1 regulates proliferation, apoptosis, angiogenesis, invasion, and migration in a variety of tumors. PTEN controls the multicellular assembly by controlling the localization and interaction of β-Arrestin1 [143]. On the other hand, β-arrestin is an important regulator of the lipid-phosphatase activity of PTEN and can regulate PTEN-dependent cell proliferation and migration [144].

Qi et al. identified a new substrate of PTEN, Abi1, which is a core adaptor protein of the WAVE regulatory complex (WRC). Abi1 is a scaffolding protein within the WRC and mediates the membrane recruitment and stabilization of WRC subunits [145]. WAVE proteins play an important role in immunity, neurogenesis, embryogenesis, cancer invasion, and metastasis [146]. It was shown that Abi1 is significantly upregulated in PTEN-deficient breast cancer cells [147]. The protein phosphatase activity of PTEN is required for the dephosphorylation and degradation of Abi1, which otherwise promotes EMT in breast cancer.

## 4. miRNA and PTEN

miRNAs are short non-coding RNAs that play an important role in the regulation of their target genes and are involved in different cellular processes such as differentiation, embryonic development, and EMT. In most cases, miRNAs interact with the 3′UTR (untranslated regions) of their target mRNAs and subsequently downregulate their expressions [148]. Notably, several microRNAs (miRNAs) have been recognized as regulators of PTEN [149].

Unsurprisingly, several miRNAs have been shown to regulate EMT by targeting PTEN/PI3K pathways. For example, miRNA-410 was shown to induce EMT and radioresistance in lung cancer cells by targeting the PI3K/mTOR pathway [150]. Another miRNA, miR-21 post-transcriptionally down-regulates the expression of PTEN and stimulates the invasiveness of non-small cell lung cancer cells [151]. In recent years, different miRNA regulators of PI3K/AKT/PTEN pathways have been discovered in various cancers. miRNA-221/222, miRNA-20a, and miRNA-200c were reported to target PTEN and modulate metastases in ovarian and breast cancer stem cells [152,153]. In addition, a number of microRNAs including miRNA-106b, miRNA-93, miRNA-144, and miRNA-202-5p regulate cell proliferation by targeting PTEN in breast cancer [154,155,156]. A list of various miRNAs is shown in Table 1. More information about the role of miRNAs in the regulation of the expression and activity of PTEN is available in the review by Ghafouri-Fard et al. [157].

## 5. lncRNA and PTEN

A number of recent publications have demonstrated the role of long-non-coding RNA (lncRNA) as important regulators of various cellular processes. Those RNAs are transcribed longer than 200 nucleotides but not translated into functional proteins. lncRNAs can regulate gene expression at different levels: chromatin organization, transcription, mRNA stability, miRNA sponging, and post-translation modifications of proteins. Moreover, they are involved in the formation of nuclear condensates and non-membrane organelles [174].

In this respect, the expression of a lncRNA named GAEA (Glucose Aroused for EMT Activation), which is induced by a variety of stimuli including high glucose, TGF-β, CTGF, SHH, and IL-6, was shown to enhance the enzymatic activity of the MEX3C RNA-binding E3 ligase that mediates the K27-linked polyubiquitination of PTEN. This covalent modification can change the specificity of PTEN activity from phosphatidylinositol/tyrosine phosphatase to serine/threonine phosphatase. Polyubiquitinated PTEN can dephosphorylate TWIST1, SNAIL, and YAP1, thereby mediating an accumulation of EMT transcription regulators [175].

Another lncRNA, Linc00702, is usually under-expressed in colorectal cancer. Importantly, the overexpression of Linc00702 reduced the invasion and migration of colorectal cancer cells. Mechanistically, Linc00702 induces PTEN expression and thus represses the PI3K/AKT pathway, thereby playing a tumor suppressive role in cancer [176]. Linc00702 is also downregulated in non-small cell lung carcinoma. The overexpression of this lncRNA in lung cancer cell lines inhibits cell growth and invasiveness by sponging miR-510, which targets PTEN transcripts [177].

On the contrary, Yan et al. [178] showed that Linc00470 mediated the degradation of PTEN mRNA in gastric cancer. Moreover, the expression of Linc00470 correlated with a poor prognosis and distant metastases in patients with gastric cancer. Thus, it can be concluded that lncRNAs differentially affect PTEN depending on their molecular targets and the cellular context and hence have different effects on the EMT and metastasis formation.

## 6. Conclusions

Here, we attempted to summarize the current knowledge regarding the role of PTEN in the EMT process. PTEN, being the major inhibitor of the PI3K/AKT pathway, plays an important part in the regulation of cell proliferation, differentiation, metabolism, and migration. The coordination of different signaling pathways, including PI3K/AKT, TGF-β/Smad, Notch, and NF-κB, is required to control such cellular processes as EMT. The molecular particularities of crosstalk between these signaling pathways were not fully explored. In this review, we provided evidence suggesting that PTEN is involved in the regulation of EMT. Specifically, we found that the expression of PTEN is regulated by EMT transcription factors. In turn, PTEN regulates the activity of different signaling pathways that orchestrate the EMT and metastasis formation in cancer. Moreover, we referred to recent works demonstrating the influence of PTEN-targeting miRNAs and lncRNAs on the EMT.

To conclude, we would like to set the following questions that need to be answered in the future to decipher the molecular mechanism of PTEN participation in the EMT process:-Does complete ablation of PTEN or loss of its enzymatic function activate EMT on its own, or does this process require additional signaling?-Is PTEN required for the reverse process of EMT, mesenchymal-to-epithelial transition?-Can PTEN be a therapeutic target for the treatment of metastatic cancer?

Mutations in the tumor suppressor PTEN occur in different tumors. Understanding the role of PTEN in cancer invasion and in the development of metastasis provides new insights for drug development and in the search for promising new potential biomarkers. PTEN plays a crucial role in the activity of cancer stem cells, which are responsible for resistance to therapy and tumor metastatic spread [179].

Importantly, recent studies show that PTEN directly regulates resistance to radiation, chemo-, and immunotherapy [180]. The link between EMT and resistance to anticancer therapy has been reported in several reviews [10,181,182]. The general mechanism underlying EMT-driven drug resistance is related to an increased drug efflux by multiple cell membrane transporter proteins, especially the ABC transporter family of proteins, avoiding drug-induced apoptosis, slow cell proliferation, and the altered expression of molecules involved in immunosuppression [10,11]. Though cancer metastasis is the major cause of death from cancer, it is hard to overstate the problems caused by resistance to anticancer drugs. Thus, novel therapeutic strategies to activate PTEN in cancer cells may yield a more efficient way to treat drug-resistant cancers.

## Figures and Tables

**Figure 1 cancers-14-03786-f001:**
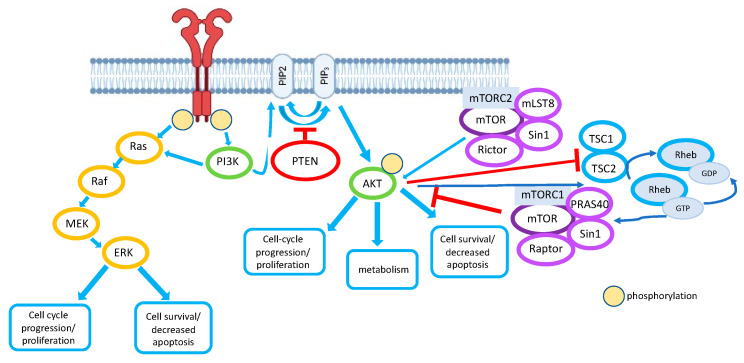
Schematic of the canonical PI3K/AKT signaling pathway. Light blue arrows denote the activating signal transduction events between the members of a particular pathway (shown as circles of different color). Red crossbars denote the inhibitory events. Upon activation of RTK (receptor tyrosine kinase), PI3K (phosphatidylinositol 3-kinase) phosphorylates and produces PIP3 (phosphatidylinositol (3,4,5)) from PIP2 (phosphatidylinositol (4,5) bisphosphate). PTEN (phosphatase) promotes the reverse reaction thus negatively regulates the pathway. PIP3 facilitates the phosphorylation of AKT (protein kinase B). Activating AKT phosphorylates different target proteins and mediates multiple cellular events and processes such as cell cycle, metabolism, survival, apoptosis, etc. Schematic representation of an alternative non-AKT-dependent activation pathway via the Ras/MAPK signaling. Upon exposure of cancer cells to extracellular signals sensed by RTKs, the signal is transferred via the Grb2/SOS protein complex to specific receptor(s) rat sarcoma virus (Ras), which binds GTP and recruits a protein kinase, rapidly accelerated fibrosarcoma (Raf). The latter relays the signal to another kinase, mitogen-activated protein kinase kinase (MEK), followed by the transfer to yet another kinase, extracellular signal-regulated kinase (ERK). ERK mediates a variety of cellular events via phosphorylation of a number of transcription factors (cell cycle, survival, apoptosis, etc.). AKT phosphorylates TSC2, which forms a functional complex with TSC1. Phosphorylation of TSC2 impairs the ability of the TSC1-TSC2 complex to act as a GAP toward the small GTPase Rheb. Rheb-GTP potently activates mTORC1. Phosphorylation of PRAS40 by Akt and by mTORC1 itself results in dissociation of PRAS40 from mTORC1 and may relieve an inhibitory constraint on mTORC1 activity. Only the key upstream phosphorylation events are shown.

**Figure 2 cancers-14-03786-f002:**
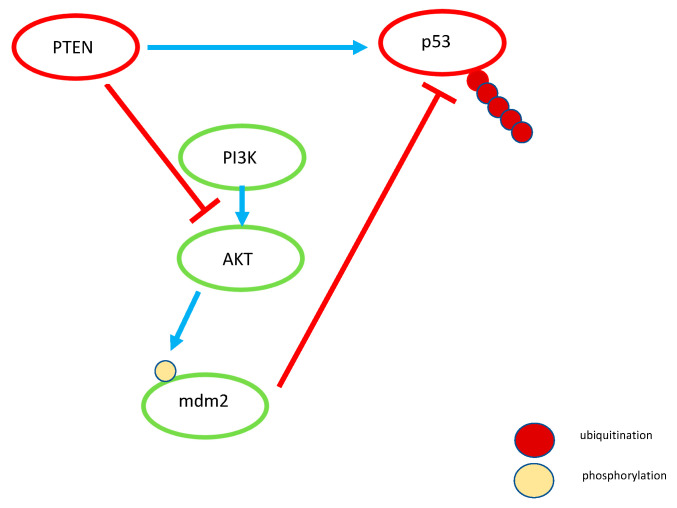
Schematic relationships between PTEN, MDM2, and p53. In addition to being the main negative regulator of the PI3K/AKT pathway, PTEN also regulates stability and transcriptional activity of p53 (tumor suppressor). Moreover, PTEN itself is a transcriptional target of p53. Another transcriptional target of p53 is a E3-ubiquitin ligase, MDM2. Importantly, MDM2 mediates the proteasomal degradation of p53. In addition, MDM2 also mediates autoubiquitination thus rendering itself an unstable protein. AKT can phosphorylate MDM2 thereby stabilizing the latter and promoting the degradation of the tumor suppressor p53. Expression of the constitutively active form of AKT augments the nuclear entry of MDM2. Light blue arrows denote activating transduction events between the members of a particular pathway (shown as circles of different colors). Red crossbars denote the inhibitory events.

**Figure 3 cancers-14-03786-f003:**
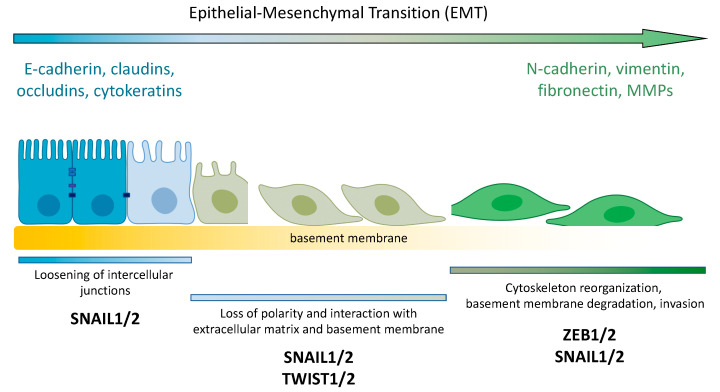
Schematic diagram of the epithelial–mesenchymal transition (EMT) and EMT transcription factors. There are a limited number of critical transcription factors that activate EMT. This core set of EMT master regulators includes zinc-finger E-box-binding homeobox 1 (ZEB1) and ZEB2, Snail (SNAI1), Slug (SNAI2), and Twist-related protein 1 (TWIST1). The EMT is a process by which epithelial cells lose their apicobasal polarity and intercellular contacts and transform into mesenchymal cells, acquiring the ability to migrate. The process of EMT causes suppressed expression of epithelial markers (such as E-cadherin) and upregulation of mesenchymal markers (such as N-cadherin). As a result, cells acquire motility and invasive properties.

**Figure 4 cancers-14-03786-f004:**
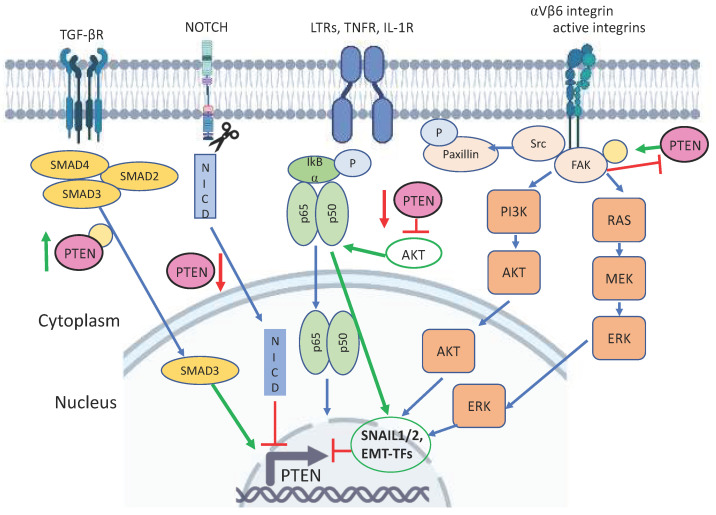
Positive and negative effects of PTEN on TGF-β, Notch, TNF, and integrin signaling pathways. Upon binding of TGF-β, the TGF-β receptor (TGF-βR) induces the phosphorylation of and nuclear translocation of SMADs, while the latter regulate transcription of EMT factors. TGF-β may potentially increase the protein level of PTEN via its phosphorylation. Notch signaling also regulates the expression of PTEN. The cleaved part of Notch receptors affects the transcription of different genes, including PTEN, which is a direct transcriptional target of Notch3. Notch1 downregulates PTEN expression. The NF-κB (Nuclear factor-κB) pathway is activated via LTRs, TNF receptors, and receptors for cytokines (e.g., IL-1R). The NFκB family consists of five subunits: Rel (c-Rel), p65 (RelA, NFκB3), RelB, p50 (NFκB1), and p52 (NFκB2). AKT-dependent activation of NF-κB induces the expression of Snail, which in turn inhibits the expression of PTEN, thereby forming the PTEN/Akt/NF-κB/Snail feedback loop. Integrins are involved in the regulation of cell adhesion to an extracellular matrix. The expression of PTEN inhibits migration and invasion through the αVβ6 integrin signaling pathway. There is a crosstalk between PTEN and FAK. On the one hand, PTEN interacts with and de-phosphorylates FAK tyrosine kinase thereby inhibiting the integrin-mediated migration and invasion of cancer cells. On the other hand, FAK phosphorylates PTEN at Tyr336, which increases the phosphatase activity and protein stability of PTEN. Interactions of PTEN with FAK and paxillin, respectively, weaken their interactions with ECM components. Thus, PTEN may contribute to the process of destabilization of cell adhesion to ECM.

**Table 1 cancers-14-03786-t001:** miRNAs targeting PTEN transcripts.

miRs	Interaction with PTEN	Role in EMT Process
miR-21	downregulation of PTEN	promotes EMT progression in lung epithelial cells [158]
miR-410	downregulation of PTEN	activation of EMT in non-small cell lung cancer [150]
miR-106b and miR-93	inhibits PTEN	promotes cell migration, invasion, and proliferation in vitro (in breast cancer cells) and tumor growth in vivo (breast cancer) [155]
miR-202-5p	downregulation of PTEN	increases DOX resistance and cell proliferation as well as inhibiting apoptosis (in breast cancer cells) [156]
miR-23b-3p	silence of PTEN	promotes EMT and migration in bronchial epithelium (lung) [159]
miR-499a-5p	decreases the expression levels of PTEN mRNA and protein	promotes 5-FU resistance and cell proliferation and migration in pancreatic cancer [160]
miR-29b	targeting the 3′-UTR of PTEN mRNA upregulation of PTEN and downregulation of PI3K in cervical cancer cells	decreases cell proliferation, migration, and invasion abilities of NSCLC cells [161]
miR-148a, miR-152 and miR-200b	downregulation of PTEN	presence of these miRs correlates with metastasis in patients with prostate cancer and metastatic prostate cancer [162]
miR-454-3p	suppresses PTEN	promotes oxaliplatin resistance and inhibits apoptosis in Colorectal cancer cell line [163]
miR-4310	suppresses PTEN SP1 activates miR-4310 gene expression	promotes proliferation, migration, and invasion in glioma tissues [164]
miR-513b-5p	decreases the level of PTEN	migration and invasion in cervical cancer tissues and cell lines [165]
miR-144-3p	inhibition of miR-144-3p expression can up-regulate PTEN	induces cell proliferation and invasion, and reduces apoptosis, in thyroid cancer [166]
miR-19	inhibits PTEN mRNA expression	decreases the expression of E-cadherin and increases the expression of α-SMA and fibronectin, while inhibition of miR-19 reverses TGF-β1-induced EMT in renal tubular epithelial cells, thereby promoting renal fibrosis [167]
miR-106b	suppresses the expression of PTEN	induces EMT in esophageal squamous cell carcinoma [168]
miR-17-5p	inhibits PTEN	enchases survival of breast cancer cells [169]
miR-552-5p	inhibits PTEN	promotes proliferation and migration and inhibits apoptosis in gastric cancer cells and stimulates metastasis in vivo; upregulation of miR-552-5p led to an increase in N-cadherin and vimentin and a reduction in E-cadherin [170]
miR-616-3p	reduced the mRNA and protein levels of PTEN	promotes proliferation and migration of endometrial stromal cells [171]
miR-181a	inhibits PTEN	promotes the proliferation and metastasis of Hepatocellular carcinoma cells in vitro and in vivo [172]
miR-221	regulates PTEN	promotes invasion and metastasis in Extrahepatic cholangiocarcinoma [173]

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
