# Peer review of "The Role of PTEN in Epithelial–Mesenchymal Transition"

_cancers, 2022, doi:10.3390/cancers14153786_

Round 1

Reviewer 1 Report

This review is a concise summary of a very big and sometimes confusing field. It should operate as an access point for the reader to the relevant literature.

I think more [and better] diagrams are needed to explain confusing feedback and cross-talk relationships between key components in the EMT and PI3K/PTEN signalling networks; this would allow easier reading and understanding of the text. My suggestions are to add clear diagrams illustrating:

The relationship between PI3K, Akt, mTORC1, mTORC2, GSK3b and growth/autophagy, apoptosis/survival.

The relative roles of PTEN as a lipid phosphatase, protein phosphatase and non-enzymic functions and how these relate to its proposed roles in the cytosol and nucleus.

The relationship between PTEN, p53 and Mdm2

A clearer and more extensive diagram (or set of diagrams) linking the key pathways regulating EMT with PTEN expression and function - Fig 3 is a start but doesn't go far enough.

Author Response

This review is a concise summary of a very big and sometimes confusing field. It should operate as an access point for the reader to the relevant literature.

We are grateful to the reviewer for careful reading of the manuscript.

I think more [and better] diagrams are needed to explain confusing feedback and cross-talk relationships between key components in the EMT and PI3K/PTEN signalling networks; this would allow easier reading and understanding of the text. My suggestions are to add clear diagrams illustrating:

The relationship between PI3K, Akt, mTORC1, mTORC2, GSK3b and growth/autophagy, apoptosis/survival.

We have corrected figure 1

The relative roles of PTEN as a lipid phosphatase, protein phosphatase and non-enzymic functions and how these relate to its proposed roles in the cytosol and nucleus.

The relationship between PTEN, p53 and Mdm2

We have added figure 2

A clearer and more extensive diagram (or set of diagrams) linking the key pathways regulating EMT with PTEN expression and function - Fig 3 is a start but doesn't go far enough.

We have corrected figure 4

Reviewer 2 Report

Fedorova and collaborators focused the attention on PTEN and its role in the regulation of biochemical pathways involved in the EMT process. The review is clearly written and the reader has an overall view of the different functions of PTEN.

My suggestion is to add a short paragraph or to improve the conclusion in order to better underline the last findings about PTEN and cancer stem cells (for instance doi: 10.1038/s41598-020-69698-, doi: 10.3390/cancers11081076) and PTEN in therapy resistance (doi: 10.1186/s13578-022-00778-7). This is to highlight the importance of PTEN pathways in the acquisition of the aggressive phenotype in the tumor.

Authors should also add this recent reference (doi: 10.1016/j.biopha.2020.110986) in the paragraph of regulatory microRNAs.

Author Response

Fedorova and collaborators focused the attention on PTEN and its role in the regulation of biochemical pathways involved in the EMT process. The review is clearly written and the reader has an overall view of the different functions of PTEN.

We appreciate the reviewer for the positive evaluation of our manuscript.

My suggestion is to add a short paragraph or to improve the conclusion in order to better underline the last findings about PTEN and cancer stem cells (for instance doi: 10.1038/s41598-020-69698-, doi: 10.3390/cancers11081076) and PTEN in therapy resistance (doi: 10.1186/s13578-022-00778-7). This is to highlight the importance of PTEN pathways in the acquisition of the aggressive phenotype in the tumor.

We have added paragraph about PTEN and cancer stem cells and PTEN in therapy resistance

Authors should also add this recent reference (doi: 10.1016/j.biopha.2020.110986) in the paragraph of regulatory microRNAs.

We have added reference

Round 2

Reviewer 1 Report

The manuscript has been improved

Minor comments:

Fig 1: the relationship between mTORC1 and Akt/PI3K is misleading (Akt activates mTORC1 through phosphorylation of TSC2 and PRAS40).

The selective use of the phosphorylation symbol in this and other figures is also a bit confusing; the authors should make it clear that only key upstream phosphorylations are shown.

I'm not sure what the isolated PTEN symbol represents in Fig 2

Author Response

Authors' Responses to Reviewer's Comments (Reviewer 1)

We are grateful to the reviewer for careful reading of the manuscript.

Minor comments:

Fig 1: the relationship between mTORC1 and Akt/PI3K is misleading (Akt activates mTORC1 through phosphorylation of TSC2 and PRAS40).

The selective use of the phosphorylation symbol in this and other figures is also a bit confusing; the authors should make it clear that only key upstream phosphorylations are shown.

We have corrected Figure 1 and added the relationship between mTORC1 and PI3K/AKT. The explanation to this connection as well as to the key phosphorylation events has been added to the figure legend (figure 1).

I'm not sure what the isolated PTEN symbol represents in Fig 2

We corrected figure 2.